# Effect of Lipid-Testing Interval on Stoke Risk among Newly Diagnosed Dyslipidemia Patients Initiated on Statins

**DOI:** 10.3390/jcm8050742

**Published:** 2019-05-24

**Authors:** Ahryoung Ko, Seulggie Choi, Jooyoung Chang, Sang Min Park

**Affiliations:** 1Department of Family Medicine, Seoul National University Hospital, Seoul National University College of Medicine, Seoul 03080, Korea; ahryoungko@gmail.com; 2Department of Biomedical Sciences, Seoul National University Graduate School, Seoul 03082, Korea; seulggie@gmail.com (S.C.); joomyjoo@gmail.com (J.C.)

**Keywords:** dyslipidemia, lipid-testing interval, stroke, statin

## Abstract

(1) Background: Although current guidelines recommend regular lipid testing for dyslipidemia patients, the effectiveness of regular lipid profile monitoring in clinical outcomes is unclear. (2) Methods: We assessed 64,664 newly diagnosed dyslipidemia patients from the Korean National Health Insurance Service Health Screening Cohort from 2003–2011 For lipid-testing frequency from all admission and outpatient records for 3 years after diagnosis. Participants were followed until 31 December 2015 for stroke. We used Cox regression analysis to determine the adjusted hazard ratio (aHR) for stroke according to lipid-testing interval. (3) Results: Compared to patients with lipid-testing intervals of ≤6 months, patients with >6 to ≤12 (aHR 1.32, 95% confidence interval (CI) 1.08–1.61), >12 to ≤18 (aHR 1.48, 95% CI 1.20–1.82), and >18 (aHR 1.54, 95% CI 1.25–1.90) month testing intervals had elevated risk of total stroke (*p* for trend <0.001). A significant association existed between lipid-testing interval and total and ischemic stroke risk in the >6 to ≤12 (aHR 1.62, 95% CI 1.19–2.21), >12 to ≤18 (aHR 1.87, 95% CI 1.36–2.58), and >18 (aHR 1.79, 95% CI 1.30–2.48) month interval groups, but no significant association existed between lipid-testing interval and hemorrhagic stroke risk. (4) Conclusions: Lipid-testing intervals of more than 6 months may lead to increased stroke risk among newly diagnosed dyslipidemia patients after initiation of statin treatment. Lipid testing every 6 months can lower stroke risk among dyslipidemia patients.

## 1. Introduction

Statin therapy has become the most important advancement in stroke prevention since the introduction of aspirin and blood pressure-lowering therapies. Statins not only lower the overall risk of stroke but also slow the progression of carotid atherosclerosis [1], reduce inflammation and endothelial dysfunction, decrease platelet aggregation to improve fibrinolysis, lower blood pressure, and decrease the risk of thromboembolic complications to the brain [1,2]. 

There is robust evidence that lipid abnormalities such as high plasma triglycerides (TG), low-density lipoprotein cholesterol (LDL-C), and decreased high-density lipoprotein cholesterol (HDL-C) levels are associated with increased stroke risk [3,4]. Drugs used for lipid-lowering therapy, such as statins, reduce the level of LDL-C and triglycerides levels, and increase HDL-C through modulation of cholesterol ester transfer protein [5,6]. Regular lipid profile follow-up is needed to check for an adequate patient response to lipid-lowering therapy and evaluate whether dyslipidemia is well-managed [7,8].

Lipid measurements are essential for calculating an individual’s risk of atherosclerotic cardiovascular disease (ASCVD), which includes stroke in addition to coronary heart disease and peripheral arterial disease, and determine when to initiate, adjust, or change lipid-lowering therapy [8,9]. Despite several clinical guidelines that recommend regular monitoring of lipid profiles to assess and improve adherence, the clinical benefit of such monitoring to detect non-adherence remains unproven. Recent lipid-management guidelines favor a risk-assessment approach that emphasizes lowering ASCVD risk rather than targeting specific LDL-C levels [9,10,11]. A limited number of studies have investigated the frequency of measuring lipid levels in patients who take lipid-lowering medications and its effects on long-term health such as risk of stroke, although stroke is one of the leading causes of death and disability worldwide [12].

We aimed to determine the effect of the lipid-test interval on the risk of developing strokes among newly diagnosed dyslipidemia patients initiated on statins. The patient population was obtained from the Korean National Health Insurance Service (NHIS) database. 

## 2. Subjects and Methods

### 2.1. Data Collection

The study population was derived from the Korean NHIS database. In Korea, the NHIS provides universal healthcare to all Korean citizens, resulting in an enrollment rate of 97%. The NHIS collects data from all hospital use including inpatient and outpatient visits, pharmaceutical drug prescriptions, and national health examinations [13]. For all enrollees aged 40 years or older, the NHIS provides biannual health screening examinations, which is comprised of a self-reported questionnaire on health behavior and medical history, measurements such as height, weight, and blood pressure, and blood tests. From this data, the NHIS constructed a cohort for research purposes called the National Health Insurance Service—National Health Screening Cohort (NHIS-HEALS), which contains information on health examinations, hospital use, and drug prescriptions among men and women aged 40 years or older at 2002 and were followed-up until 2015 [14]. NHIS-HEALS also contains information on the cause of death and death date, which was merged from the Statistics Korea database. The NHIS-HEALS database has previously been used for multiple epidemiological studies and its validity is described in detail elsewhere [14].

### 2.2. Study Population

After identifying newly diagnosed dyslipidemia patients during 2003–2011, the frequency of lipid testing was observed for the next 3 years after diagnosis date. Afterwards, participants were followed-up until stroke, death, or 31 December 2015, whichever came earliest. The index date was 3 years after diagnosis date of dyslipidemia. Among 75,944 newly diagnosed dyslipidemia patients during 2003–2011 who were not diagnosed with dyslipidemia in 2002, we excluded 2245 participants with missing values on covariates (Age (*n* = 26), Smoking (*n* = 1620), Alcohol (*n* = 184), Physical activity (*n* = 361), Body mass index (*n* = 9), Systolic blood pressure (*n* = 10), Fasting serum glucose (*n* = 16), Total cholesterol (*n* = 19)). Furthermore, 8731 and 304 individuals who were diagnosed with cardiovascular disease or died before the index date were excluded, respectively. The study population consisted of 64,664 newly diagnosed dyslipidemia patients.

The Seoul National University Institutional Review Board (IRB) approved this study (IRB number: E-1803-046-928) and the requirement for informed consent was waived as the NHIS-HEALS database was constructed after anonymization according to strict confidentiality guidelines.

### 2.3. Key Variables

Dyslipidemia was defined when a participant was prescribed statin medication under the International Classification of Diseases, Tenth Revision (ICD-10) code pertaining to dyslipidemia (E78), along with at least one lipid blood test within 3 years of diagnosis. Lipid testing, which includes total cholesterol, TG, HDL-C, and LDL-C, was observed for all participants for 3 years after diagnosis of dyslipidemia. Lipid-testing interval was defined as the average interval between lipid testing during the observed 3-year span. The 3-year interval was divided into 6 months, resulting in a total of 6 half-year intervals, after which whether or not each individual had undergone lipid testing for each interval was determined. Then, the total number of intervals was divided by the cumulative number of lipid tests, resulting in the average lipid-testing interval. The study population was divided according to the average lipid-testing interval of ≤6, 6 to 12, 12 to 18, and ≥18 months.

### 2.4. Identification of Stoke

Stroke was defined as 2 or more days of hospitalization or death with the cause of death under ICD-10 codes pertaining to total stroke (I60-I69). The ICD-10 codes are in line with those used by the American Heart Association [15] Total stroke was further divided into ischemic stroke (ICD-10 code I63) and hemorrhagic stroke (ICD-10 codes I61 and I62). 

### 2.5. Statistical Analysis

For determining the differences in descriptive characteristics according to average lipid-testing interval groups, Chi squared test was used for categorical variables and analysis of variance for continuous variables (Table 1). Cox proportional hazards regression was used obtain the adjusted hazard ratios (aHRs) and 95% confidence intervals (CIs) for stroke risk according to average lipid-testing interval.

Cox regression and competing-risks survival regression based on Fine and Gray’s model, were conducted to estimate the hazard ratios of stroke (Table 2). Also, the risk of coronary heart disease and cardiovascular disease mortality risk was determined according to lipid-testing frequency.

The assumption of proportionality for the Cox regression analysis was graphically tested and verified using the Schoenfeld residual method. The considered covariates include age (continuous, years), sex (categorical, men and women), household income (categorical, 1st, 2nd, 3rd, and 4th quartiles), smoking (categorical, never, past, and current smokers), alcohol consumption (categorical, none, <1, 1–2, 3–4, and ≥5 times per week), physical activity (categorical, none, 1–2, 3–4, 5–6, and 7 times per week), body mass index (continuous, kg/m^2^), systolic blood pressure (continuous, mmHg), fasting serum glucose (continuous, mg/dL), total cholesterol (continuous, mg/dL), Charlson comorbidity index (continuous), statin adherence (continuous, medication possession ratio), outpatient department visits (OPD) (categorical, 1st, 2nd, 3rd, and 4th quartiles), and enrollment year (continuous, year). Household income was determined by the insurance premium and body mass index by dividing the weight in kilograms by height in meters squared. Statin adherence was measured by medication possession ratio (MPR), which is calculated by dividing the defined daily dose (DDD) between the first and last prescription or to the end of follow-up during the 3 years after diagnosis of dyslipidemia. Statin dosage, standardized by DDD, was calculated according to the Anatomical Therapeutic Chemical classification system of drugs by the World Health Organization Collaborating Center for Drug Statistics Methodology [16]. Enrollment year was defined as the year of dyslipidemia diagnosis. Stratified analyses for total stroke risk according to subgroups of age, smoking, physical activity, alcohol consumption, total cholesterol, body mass index, outpatient department visits, and statin adherence were conducted.

Statistical significance was defined as a *p* value of less than 0.05 in a 2-sided manner. All data collection and statistical analyses were conducted using SAS 9.4 (SAS Institute Inc, Cary, NC, USA).

## 3. Results

Table 1 depicts the descriptive characteristics of the study population. The number of participants with average lipid-testing intervals of ≤6, 6 to 12, 12 to 18, and ≥18 months are 4470, 28,189, 15,727, and 16,278, respectively. The mean (standard deviation, SD) age for participants with average lipid-testing intervals of ≤6, 6 to 12, 12 to 18, and ≥18 months are 60.2 (8.1), 60.4 (8.2), 60.6 (8.5), and 60.2 (8.5) years, respectively. Compared to those with average lipid-testing interval of ≤6 months, those with an average lipid-testing interval of ≥18 months tended to have lower household income, be current smokers, exercise less, consume less alcohol, visit the outpatient department less frequently, and have less comorbid conditions (all *p* < 0.001).

Results from the effect of lipid-testing interval on stroke risk are shown in Table 2. Compared to those with lipid-testing interval of ≤6 months, participants with 6 to 12 (aHR 1.32, 95% CI 1.08–1.61), 12 to 18 (aHR 1.48, 95% CI 1.20–1.82), and ≥18 months (aHR 1.54, 95% CI 1.25–1.90) had elevated risk of total strokes. Participants with lipid-testing interval of 6 to 12 (aHR 1.62, 95% CI 1.19–2.21), 12 to 18 (aHR 1.87, 95% CI 1.36–2.58), and ≥18 months (aHR 1.79, 95% CI 1.30–2.48) had elevated risk of ischemic stroke compared to those with lipid-testing interval of ≤ 6 months. No significant association was found between lipid-testing interval and hemorrhagic stroke risk (6 to 12 (aHR 0.61, 95% CI 0.38–1.00), 12 to 18 (aHR 0.72, 95% CI 0.43–1.21), and ≥18 months (aHR 0.85, 95% CI 0.50–1.42).

Table 3 and Table A1 depict the results from stratified analyses on the effect of average lipid-testing interval on total strokes, respectively. The positive association between lipid-testing interval and total strokes was preserved among subgroups of sex, drug adherence and outpatient department visits (Table 3). The increased risk effect for stroke with longer lipid-testing intervals tended to be more prevalent among those who were within aged ≥60 years (Table 3, *p* < 0.05), past or current smokers, exercised, did not consume alcohol, total cholesterol <240 mg/dL and had body mass index <25 kg/m^2^ (Table A1, all *p* < 0.05). Finally, the risk of coronary heart disease or cardiovascular disease mortality was not increased upon greater lipid-testing intervals (Table A2).

## 4. Discussion

In this retrospective cohort study, we showed that an association existed between increased lipid-testing intervals of more than 6 months and elevated risk of total stroke among newly diagnosed dyslipidemia patients initiated on statins. Compared to those with lipids tested at <6 months intervals, those patients tested every 6–12 months, 12–18 months and >18 months had higher rates of ischemic stroke event rates over the next three 6-month intervals compared to event rates in the first 6 months. There was no significant increase in hemorrhagic strokes, but when we combined the stroke diagnoses, there was a significant combined increase over 6 months with less average lipid screening. To the best of our knowledge, this was the first study to show an elevated the risk of stroke when the interval of lipid testing was longer than 6 months among patients who initiated statin treatment.

There is insufficient evidence on the stroke risk of those who do not undergo regular lipid testing after initiating lipid-lowering medications, despite the high prevalence of lipid testing in the clinical practice [17]. Lipid-lowering management with statins and lipid testing is important to improve or optimize the clinical status of patients. A previous study has suggested that because the true underlying cholesterol levels change slowly, clinicians may monitor cholesterol too often to evaluate real trends of lipid profiles [18]. Guidelines recommend that clinicians monitor patients’ cholesterol levels to assess adherence to lipid-lowering medications or assess the risk of ASCVD by reviewing their lipid profiles [9,19,20]. For patients, blood cholesterol measurements may provide motivation to adhere to lipid profile management, including adherence to medications and modifications of lifestyles [21]. In this study, we have shown an association between longer lipid-testing intervals of more than 6 months and elevated risk of stroke, even after adjustments for a wide range of potential confounders such as drug adherence, OPD visits, and health behaviors that could impact lipid modifications.

Previous studies suggested that frequent and regular lipid monitoring played an important role in long-term adherence [22,23]. Short-term follow-up with lipid testing after initiation of statins might be beneficial in cases of poor compliance to therapy. Patients who visit hospitals frequently are more likely to have the chance to consult with their physicians. Thus, OPD visits may act as a surrogate marker for a chance for a therapeutic intervention through physician follow-up. Management by physicians is crucial for optimizing adherence to medications and health behaviors that include smoking habit, regular exercise, diet, and alcohol consumption [24]. Therefore, we conducted stratified analyses in this study according to the subgroups of drug adherence and OPD visits. Several mechanisms might explain the risk-increasing effect for stroke according to the increasing lipid-testing intervals in our study. First, the lack of lipid testing might lead to inappropriate adjustment of lipid-lowering therapy because the lipid profiles are unknown. Possibly, a patient has a high drug adherence without monitoring laboratory results and, lipid management care might be suboptimal because the patient could be undertreated but a less intensive lipid management and would not achieve target LDL-C levels. Second, lack of awareness of lipid levels might lead to inappropriate risk-assessment and lifestyle intervention. A physician would need to control the ASCVD risk of a patient according to the result of the lipid laboratory test. If the test is not conducted, a physician would have less chance to educate, emphasize, or modify the patient’s health behaviors. Modifiable risk factors of stroke such as elevated blood pressure, serum glucose, cholesterol levels and obesity could be prevented and controlled by offering lifestyle changes or explaining the benefits of a healthy lifestyle to dyslipidemia patients [25,26]. Third, knowledge of the cholesterol level may be important to motivate patients to talk with physicians and change behaviors to improve their cholesterol levels. Even though a randomized trial of the motivational effect of cholesterol measurement in general practice has shown only negligible benefit [27], a recent trial of diabetes patients randomized to self-monitoring their risk factors showed better achievement of target measurements of blood pressure, LDL-C, and HbA1c, as well as a reduction in clinical events [21]. Other studies of type 2 diabetes and hypertension patients have showed that well-informed and motivated patients were more insistent to reach and maintain target values and, ultimately reduce the risk of stroke [28,29]. Likewise, knowing their cholesterol levels would motivate and encourage patients to comply with lipid-lowering therapy to improve their dyslipidemia and cardiovascular health.

As pointed out by other well-conducted studies, besides the anti-stroke effect of lipid-lowering agents, multi-factor risk factors should also be taken into consideration including stroke subtype [30,31]. Only the risk of ischemic stroke increased with increasing lipid-test interval, but the effect for the risk of hemorrhagic stroke was not evident in our study. First, our expectation from previous studies is that the incidence of obstructive vascular disease is proportional to the normal LDL-C concentration. In the Multiple Risk Factor Intervention Trial (MRFIT), which investigated the association between cholesterol and other underlying pathologic types of strokes and The Copenhagen City Heart Study (CCHS) had a similar dose-response relationship between serum cholesterol and ischemic stroke risk, but the risk of cholesterol and hemorrhagic stroke was reversed [32,33]. Second, our results can be expected from the result that statins reduce ischemic stroke and do not affect cerebral hemorrhage. Many recent studies have focused on the effect of statins on the development of stroke subtypes, and reported reductions in the risk of ischemic stroke and a significant increase in the risk of hemorrhagic stroke [34]. Therefore, the results of this study have suggested that intensive lipid profile management and correction of risk factors for stroke are necessary to improve clinical outcome. Careful consideration is needed when interpreting the exact reason for the inconsistency of ischemic stroke and bleeding risk according to the interval of lipid testing, and should be investigated further.

Several limitations must be considered when interpreting the results from our study. First, there could be an inaccuracy of the information in defining newly diagnosed dyslipidemia patients due to the nature of the claims database. However, previous studies for the diagnosis codes of the NHIS data have shown that 70% of the data from the NHIS database matched with those from patients’ medical records [35,36]. Second, we could not account for LDL –C levels due to the lack of information—not only the test for LDL-C, but also total cholesterol, TG and HDL-C levels not included in this study. Although high LDL-C levels are necessary to prescribe lipid-lowering medications [37], total cholesterol, plasma TG level, and HDL-C level have previously been shown to be associated with several adverse outcomes [38,39,40]. Regarding the review of lipid abnormality with elevated total cholesterol, plasma TG, and decreased HDL-C concentrations, our study investigated the effect of monitoring lipid profiles in patients initiating statins. Third, we could not determine if dyslipidemia patients who underwent undergone stable lipid therapy or achieved treatment goals after initiating statins was due to the lack of lipid levels. Therefore, future studies that differentiate patients should be conducted to determine the effect of the testing interval on stroke event between the two groups of before and after the achieved treatment goal [41]. Fourth, we did not consider the effect of lipid monitoring on stroke risk beyond 3 years after initial diagnosis. The early pattern of lipid monitoring might represent the patients’ general tendencies. These results also emphasized the importance of early frequent follow-up with lipid testing. Finally, our results given in the manuscript showing that lipid-testing frequency was not associated with higher risk of coronary heart disease or cardiovascular disease mortality. These results imply that intensive lipid profile management and modification of major risk factor for strokes is required for improving clinical outcome. On the other hand, CHD risk may have sufficiently decreased by taking statin medications, resulting in a weaker protective effect for CHD upon frequent lipid testing. It remains cautious in interpreting our results the exact reasons for the discrepancy in total stroke and CHD risk according to lipid-testing interval and merit further investigation.

The association of lipid-testing interval with coronary artery disease or CVD mortality, however, could not be assessed properly and thus merit future studies

## 5. Conclusions

This is the first study of a relatively large population to report the elevated risk of stroke according to longer lipid-testing intervals among patients who initiated statins therapy. These results suggest that longer lipid-testing intervals of more than 6 months might lead to elevated risk of stroke. Dyslipidemia patients who begun statins should be monitored for lipid levels to benefit from reduced risk of ASCVD. Our findings have also supported the 6 months interval for lipid testing among newly diagnosed dyslipidemia patients who initiated lipid-lowering drugs suggested by current guidelines for management [10]. This study also provides the grounds for physicians to assess appropriate ASCVD risk and lifestyle intervention to dyslipidemia patients and emphasize the need for strict dyslipidemia control.

## Figures and Tables

**Table 1 jcm-08-00742-t001:** Descriptive characteristics of the study population.

	Average Lipid-Testing Interval, Months				
	≤6	6 to 12	12 to 18	≥18	*p* Value
Number of people	4470	28,189	15,727	16,278	
Age, years, mean (SD)	60.2 (8.1)	60.4 (8.2)	60.6 (8.5)	60.2 (8.5)	<0.001
Sex, N (%)					
Men	2046 (45.8)	11,233 (39.9)	6596 (41.9)	7356 (45.2)	<0.001
Women	2424 (54.2)	16,956 (60.2)	9131 (58.1)	8922 (54.8)	
Household income, quartiles, N (%)					
1st (highest)	1872 (41.9)	10,044 (35.6)	5433 (34.6)	5552 (34.1)	<0.001
2nd	1260 (28.2)	8379 (29.7)	4726 (30.1)	4887 (30.0)	
3rd	800 (17.9)	5525 (19.6)	3241 (20.6)	3369 (20.7)	
4th (lowest)	538 (12.0)	4241 (15.0)	2327 (14.8)	2470 (15.2)	
Smoking status, N (%)					
Never smoker	3245 (72.6)	21,209 (75.2)	11,601 (73.8)	11,570 (71.1)	<0.001
Past smoker	787 (17.6)	3917 (13.9)	2167 (13.8)	2317 (14.2)	
Current smoker	438 (9.8)	3063 (10.9)	1959 (12.5)	2391 (14.7)	
Physical activity, times per week, N (%)					
None	2090 (46.8)	14,089 (50.0)	8161 (51.9)	8499 (52.2)	<0.001
1–2	1054 (23.6)	6480 (23.0)	3627 (23.1)	3805 (23.4)	
3–4	709 (15.9)	4177 (14.8)	2201 (14.0)	2235 (13.7)	
5–6	322 (7.2)	1730 (6.1)	877 (5.6)	891 (5.5)	
7	295 (6.6)	1713 (6.1)	861 (5.5)	848 (5.2)	
Alcohol consumption, times per week, N (%)					
None	3052 (68.3)	19,069 (67.7)	10,353 (65.8)	10,246 (62.9)	<0.001
<1	616 (13.8)	3878 (12.8)	2157 (13.7)	2506 (15.4)	
1–2	419 (9.4)	2701 (9.6)	1624 (10.3)	1750 (10.8)	
3–4	259 (5.8)	1731 (6.1)	1003 (6.4)	1189 (7.3)	
≥5	124 (2.8)	810 (2.9)	590 (3.8)	587 (3.6)	
Body mass index, kg/m^2^, mean (SD)	24.7 (3.0)	24.8 (3.0)	24.7 (2.9)	24.6 (2.9)	<0.001
Systolic blood pressure, mmHg, mean (SD)	126.1 (15.4)	126.8 (15.1)	127.2 (15.2)	126.9 (15.1)	<0.001
Fasting serum glucose, mg/dL, mean (SD)	108.0 (30.1)	105.5 (29.3)	104.0 (28.0)	103.1 (27.8)	<0.001
Total cholesterol, mg/dL, mean (SD)	189.3 (44.8)	202.7 (45.8)	209.4 (45.9)	214.6 (44.6)	<0.001
OPD visits, average interval, N (%)					
1st (mean 9 visits per year)	677 (15.2)	5008 (17.8)	4237 (26.9)	5989 (36.8)	<0.001
2nd (mean 17 visits per year)	1151 (25.8)	7368 (26.1)	4079 (25.9)	4408 (24.9)	
3rd (mean 26 visits per year)	1242 (27.8)	7519 (26.7)	3711 (23.6)	3342 (20.5)	
4th (mean 53 visits per year)	1400 (31.3)	8294 (29.4)	3700 (23.5)	2899 (17.8)	
Statin adherence, MPR, mean (SD)	0.6 (0.3)	0.5 (0.3)	0.4 (0.3)	0.3 (0.3)	<0.001
Charlson comorbidity index, N (%)					
0	310 (6.9)	2502 (8.9)	1801 (11.5)	2371 (14.6)	<0.001
1	731 (16.4)	5501 (19.5)	3683 (23.4)	4469 (27.5)	
2	938 (21.0)	6789 (24.1)	3979 (25.3)	4137 (25.4)	
≥3	2491 (55.7)	13,397 (47.5)	6264 (39.8)	5301 (32.6)	

*p* value calculated by the Chi squared test for categorical variables and analysis of variance for continuous variables. Acronyms: SD, standard deviation; OPD, outpatient department; MPR, medication possession ratio.

**Table 2 jcm-08-00742-t002:** Hazard ratios for stroke according to average lipid-testing interval among newly diagnosed dyslipidemia patients.

	Average Lipid-Testing Interval, Months				
	≤6	6 to 12	12 to 18	≥18	*p* for Trend
Total stroke					
Events	111	885	536	546	
Person-years	21,458	133,245	73,079	75,609	
aHR (95% CI)	1.00 (reference)	1.32 (1.08–1.61)	1.48 (1.20–1.82)	1.54 (1.25–1.90)	<0.001
Ischemic stroke					
Events	45	451	287	269	
Person-years	21,458	133,245	73,079	75,609	
aHR (95% CI)	1.00 (reference)	1.62 (1.19–2.21)	1.87 (1.36–2.58)	1.79 (1.30–2.48)	0.004
Hemorrhagic stroke					
Events	22	77	49	56	
Person-years	21,458	133,245	73,079	75,609	
aHR (95% CI)	1.00 (reference)	0.61 (0.38–1.00)	0.72 (0.43–1.21)	0.85 (0.50–1.42)	0.455

Hazard ratio calculated by Cox proportional hazards regression after adjustments for age, sex, household income, smoking status, physical activity, alcohol consumption, body mass index, systolic blood pressure, fasting serum glucose, total cholesterol, outpatient department visits, statin adherence, Charlson comorbidity index, and enrollment year. Acronyms: aHR, adjusted hazard ratio; CI, confidence interval.

**Table 3 jcm-08-00742-t003:** Stratified analysis on the effect of average lipid-testing interval on total stroke among newly diagnosed dyslipidemia patients according to subgroups of age, sex, outpatient department visits, and statin adherence.

	Adjusted Hazard Ratio (95% Confidence Interval)				
	Average Lipid-Testing Interval, Months				
	≤6	6 to 12	12 to 18	≥18	*p* for Trend
Age					
<60 years	1.00 (reference)	1.19 (0.85–1.67)	1.41 (0.99–2.02)	1.36 (0.94–1.96)	0.050
≥60 years	1.00 (reference)	1.43 (1.12–1.83)	1.64 (1.27–2.12)	1.81 (1.40–2.34)	<0.001
Sex					
Men	1.00 (reference)	1.36 (1.00–1.84)	1.51 (1.10–2.08)	1.59 (1.16–2.19)	0.004
Women	1.00 (reference)	1.28 (0.99–1.67)	1.44 (1.09–1.90)	1.50 (1.13–1.98)	0.003
OPD visits					
Upper half	1.00 (reference)	1.09 (0.76–1.55)	1.12 (0.78–1.60)	1.28 (0.92–1.79)	0.031
Lower half	1.00 (reference)	1.17 (0.91–1.49)	1.17 (0.91–1.52)	1.29 (1.01–1.64)	0.040
Statin adherence					
MPR<0.5	1.00 (reference)	1.09 (0.84–1.42)	1.04 (0.79–1.36)	1.23 (0.96–1.59)	0.041
MPR≥0.5	1.00 (reference)	1.21 (0.88–1.66)	1.34 (0.97–1.85)	1.39 (1.03–1.89)	0.022

Hazard ratio calculated by Cox proportional hazards regression after adjustments for age, sex, household income, smoking status, physical activity, alcohol consumption, body mass index, systolic blood pressure, fasting serum glucose, total cholesterol, outpatient department visits, statin adherence, Charlson comorbidity index, and enrollment year. Acronyms: OPD, outpatient department; MPR, medication possession ratio.

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
