# Peer review of "Effect of Lipid-Testing Interval on Stoke Risk among Newly Diagnosed Dyslipidemia Patients Initiated on Statins"

_jcm, 2019, doi:10.3390/jcm8050742_

Round 1
Reviewer 1 Report
The subject of the article is interesting and important, and the number of patients are bigger 66,146
General comments:
1- There are no tables, then it is impossible to analyse the results.
2- The data collection must be improved and clarified, and the used of statin must be better explored.
3- The references to the tables should be eliminated from the discussion.
Author Response
Response to comments from Reviewer 1
1. There are no tables, then it is impossible to analyse the results.
We would like to thank the reviewer for this comment. In the revised manuscript, we have placed the Tables (Table 1, Table 2 and Table3) in the main text near to the first time they are cited. Supplemental tables (Table S1 and Table S2) were in the appendix section.
2. The data collection must be improved and clarified, and the used of statin must be better explored.
We appreciate this fine comment, which was helpful in improving our manuscript. We agree that additional explanation of the study design and data collection. Moreover, in response to Reviewer 2’s comment 1 and 7, we have also added details of missing variables and statin dosage.
Page 4 lines 100-106:
Among 75,944 newly diagnosed dyslipidemia patients during 2003-2011 who were not diagnosed with dyslipidemia in 2002, we excluded 2,245 participants with missing values on covariates (Age (n=26), Smoking (n=1,620), Alcohol (n=184), Physical activity (n=361), Body mass index (n=9), Systolic blood pressure (n=10), Fasting serum glucose (n=16), Total cholesterol (n=19)). Furthermore, 8,731 and 304 individuals who were diagnosed with cardiovascular disease or died before the index date were excluded, respectively. The study population consisted of 64,664 newly diagnosed dyslipidemia patients.
Page 6, lines 131-135:
“Statin adherence was measured by medication possession ratio (MPR), which is calculated by dividing the defined daily dose (DDD) between the first and last prescription or to the end of follow-up during the 3 years after diagnosis of dyslipidemia. Statin dosage, standardized by DDD, was calculated according to the Anatomical Therapeutic Chemical classification system of drugs by the World Health Organization Collaborating Center for Drug Statistics Methodology.”
3. The references to the tables should be eliminated from the discussion.
We would like to thank the reviewer for this comment, which was helpful in improving our manuscript. We have deleted the references in our discussion sections.
Reviewer 2 Report
1. The authors exclude 22693 due to missingness. Please provide comparison between those and the study population with respect to age, sex, incident disease and other available covariates to check if selective missingness was present.
2. Please provide a more detailed explanation on how the cox models were adjusted in the statistical methods section. Which variables were included in the model as continuous variables and which were stratified for?
3. Did the authors test the proportional hazards assumption in their models?
4. Tables are missing from the manuscript.
5. Any sex differences in the defined intervals?
6. What are the mean lipid levels in each of these intervals?
7. If the authors have access to statin dosages, please provide the dosages standardized to simvastatin during the first 3 year period used to define intervals.
8. The authors should consider competing risks by total and cardiovascular deaths in their analyses.
9. Have the authors considered same analyses for other incident atherosclerotic diseases and mortality, and particularly coronary artery disease and cardiovascular mortality? Does it show the same pattern?
Author Response
We are very grateful for the reviews provided by the editors and each of the external reviewers of this manuscript. The comments are encouraging and the reviewers appear to share our judgement that this study and its results are clinically important. Please see below, in blue, our detailed response to comments. All page numbers refers to the manuscript file with tracked changes.
Response to comments from Reviewer 2
1. The authors exclude 22693 due to missingness. Please provide comparison between those and the study population with respect to age, sex, incident disease and other available covariates to check if selective missingness was present.
We would like to thank the reviewer for this fine comment. In the revised manuscript, we edited the Results section accordingly.
Page 4 lines 100-106:
Among 75,944 newly diagnosed dyslipidemia patients during 2003-2011 who were not diagnosed with dyslipidemia in 2002, we excluded 2,245 participants with missing values on covariates (Age (n=26), Smoking (n=1,620), Alcohol (n=184), Physical activity (n=361), Body mass index (n=9), Systolic blood pressure (n=10), Fasting serum glucose (n=16), Total cholesterol (n=19)). Furthermore, 8,731 and 304 individuals who were diagnosed with cardiovascular disease or died before the index date were excluded, respectively. The study population consisted of 64,664 newly diagnosed dyslipidemia patients.
2. Please provide a more detailed explanation on how the cox models were adjusted in the statistical methods section. Which variables were included in the model as continuous variables and which were stratified for?
We would like to thank the reviewer for this comment.
We have described the type of variables in the ‘key variables’ in Methods. In revised manuscript, we described the explanation in the ‘statistical methods’ section accordingly.
Page 6,7 lines 149-156:
The considered covariates include age (continuous, years), sex (categorical, men and women), household income (categorical, 1st, 2nd, 3rd, and 4th quartiles), smoking (categorical, never, past, and current smokers), alcohol consumption (categorical, none,<1, 1-2, 3-4, and ≥5 times per week), physical activity (categorical, none, 1-2, 3-4, 5-6, and 7 times per week), body mass index (continuous, kg/m2), systolic blood pressure (continuous, mmHg), fasting serum glucose (continuous, mg/dL), total cholesterol (continuous, mg/dL), Charlson comorbidity index (continuous), statin adherence (continuous, medication possession ratio), outpatient department visits (OPD) (categorical, 1st, 2nd, 3rd, and 4th quartiles), and enrollment year (continuous, year)Stratified analyses for total stroke risk according to subgroups of age, smoking, physical activity, alcohol consumption, total cholesterol, body mass index, outpatient department visits, and statin adherence were conducted.
3. Did the authors test the proportional hazards assumption in their models?
We would like to thank the reviewer for this excellent comment. In the revised manuscript, we have described that the proportionality assumption was tested and verified using the Schoenfeld residual method.
Page 6, lines 148-149:
“The assumption of proportionality for the Cox regression analysis was graphically tested and verified using the Schoenfeld residual method.”
4. Tables are missing from the manuscript.
We would like to thank the reviewer for this comment. In the revised manuscript, we have placed the Tables (Table 1, Table 2 and Table3) in the main text near to the first time they are cited. Supplemental tables (Table S1 and Table S2) were in the appendix section.
5. Any sex differences in the defined intervals?
We would like to thank the reviewer for this most excellent comment, which was helpful in improving our manuscript.
Page 10, lines 189 :
We included the Stratified analyses for total stroke risk according to subgroups in the Table 3 that the positive association between lipid testing interval and total strokes was preserved among subgroups of sex, drug adherence and outpatient department visits.
6. What are the mean lipid levels in each of these intervals?
We would like to thank the reviewer for this excellent comment. We also agree that the lack of consideration of lipid levels during the lipid testing in dyslipidemia patients is an important limitation of our study and have elaborated on this in the Discussion section. we agree that future studies that use not only LDL levels but also TG and total cholesterol levels would be beneficial and have elaborated on this in the Discussion section.
Page 16, lines 286-293:
Second, we could not account for LDL –C levels due to the lack of information. Not only the test for LDL-C, but also total cholesterol, TG and HDL-C levels not included in this study. Although high LDL-C levels are necessary to prescribe lipid-lowering medications, 37 total cholesterol, plasma TG level, and HDL-C level have previously been shown to be associated with a number of adverse outcomes.38-40 Regarding the review of lipid abnormality with elevated total cholesterol, plasma TG and decreased HDL-C concentrations, our study investigated the effect of monitoring lipid profiles in patients initiating statins.
7. If the authors have access to statin dosages, please provide the dosages standardized to simvastatin during the first 3 year period used to define intervals.
We would like to thank the reviewer for this excellent comment. When determining statin medication possession ratio (MPR) values, statin dosage was considered in the defined daily dose (DDD) system. Statin DDD was calculated according to the Anatomical Therapeutic Chemical classification system of drugs by the World Health Organization Collaborating Center for Drug Statistics Methodology. Therefore, statin dosage for all types of statins were standardized into DDDs and then calculate into MPR values. We have edited the Methods section of our revised manuscript to elaborate on the DDD system.
Page 6, lines 117-122:
“Statin adherence was measured by medication possession ratio (MPR), which is calculated by dividing the defined daily dose (DDD) between the first and last prescription or to the end of follow-up during the 3 years after diagnosis of dyslipidemia. Statin dosage, standardized by DDD, was calculated according to the Anatomical Therapeutic Chemical classification system of drugs by the World Health Organization Collaborating Center for Drug Statistics Methodology.”
8. The authors should consider competing risks by total and cardiovascular deaths in their analyses.
We would like to thank the reviewer for this fine comment, which was most helpful in improving our study. Survival regression analyses, Cox regression and competing-risks survival regression based on Fine and Gray’s model, were conducted to estimate the hazard ratios of all-cause mortality and CVDs respectively. Competing-risk survival regression has been known as a useful alternative to Cox regression in the presence of competing risks. We included the Table 2 Hazard ratios for stroke according to average lipid testing interval among newly diagnosed dyslipidemia patients after consideration of competing risks
9. Have the authors considered same analyses for other incident atherosclerotic diseases and mortality, and particularly coronary artery disease and cardiovascular mortality? Does it show the same pattern?
We would like to thank the reviewer for this comment. We have conducted the analysis by dividing atherosclerotic disease to coronary heart disease(CHD) (ICD-10, I20-25) and cardiovascular mortality (Table S2). This result is not consistent with the results given in the manuscript showing that lipid testing frequency was not associated with higher risk of CHD or CVD mortality.
These results imply that intensive lipid profile management and modification of major risk factor for strokes is required for improving clinical outcome. On the other hand, CHD risk may have sufficiently decreased by taking statin medications, resulting in a weaker protective effect for CHD upon frequent lipid testing. It remains cautious in interpreting our results the exact reasons for the discrepancy in total stroke and CHD risk according to lipid testing interval and merit further investigation.
The association of lipid testing interval with coronary artery disease or CVD mortality however, could not be assessed properly and thus merit future studies.
Round 2
Reviewer 1 Report
The authors have revised the manuscript according to the reviewer's comments.
Reviewer 2 Report
The authors addressed all my concerns